# Dental Age Estimation by Demirjian, Willems, and Cameriere Methods in Children with Familial Mediterranean Fever: A Case–Control Study

**DOI:** 10.3390/children12111458

**Published:** 2025-10-27

**Authors:** Halenur Altan, Ergün Sönmezgöz, Melek Belevcikli, Nihal Altunok Ünlü, Ali Gül, Müzeyyen Dilşah Demiray, Ahmet Altan

**Affiliations:** 1Department of Pediatric Dentistry, Faculty of Dentistry, Necmettin Erbakan University, 42090 Konya, Türkiye; halenur.altan@erbakan.edu.tr; 2Department of Pediatrics, Tokat Medical Park Hospital, 60230 Tokat, Türkiye; esonmezgoz@gmail.com; 3Department of Pediatric Dentistry, Faculty of Dentistry, Bülent Ecevit University, 67600 Zonguldak, Türkiye; melek.belevcikli@beun.edu.tr; 4Department of Pediatric Dentistry, Faculty of Dentistry, Bolu Abant İzzet Baysal University, 14030 Bolu, Türkiye; nihal.altunokunlu@ibu.edu.tr; 5Department of Pediatrics, Faculty of Medicine, Gaziosmanpaşa University, 60250 Tokat, Türkiye; ali.gul@gop.edu.tr; 6Department of Oral and Maxillofacial Surgery, Faculty of Dentistry, Necmettin Erbakan University, 42090 Konya, Türkiye; ahmet.altan@erbakan.edu.tr

**Keywords:** forensic odontology, dental age estimation, familial mediterranean fever, Cameriere, Willems, Demirjian

## Abstract

**Highlights:**

**What are the main findings?**

**What are the implications of the main findings?**

**Abstract:**

**Background and objectives:** This study aimed to investigate the effects of familial Mediterranean fever (FMF)—a chronic inflammatory disease—on dental maturation regarding dental development using three dental age estimation methods. **Methods:** The orthopantomograms of 78 children diagnosed with FMF were compared with those of 78 systemically healthy control children. Demirjian, Willems, and Cameriere’s methods were used to estimate the dental age from seven teeth in the left mandible on the orthopantomograms. The data were analyzed using R statistical software. **Results:** The mean difference between dental age (DA) and chronological age (CA) using the Demirjian method was 0.646 in the control group and 0.753 in the FMF group (*p* = 0.595). For the Willems method, the mean DA versus CA difference was 0.283 in the control group and 0.322 in the FMF group (*p* = 0.835). Regarding the Cameriere method, the mean difference between DA and CA was −0.399 for the control group and −0.435 for the FMF group (*p* = 0.863), indicating no significant differences. At the ±1-year threshold, the Willems method showed the highest accuracy (69.87%), followed by the Cameriere method (66.67%), whereas the Demirjian method had the lowest accuracy (54.49%). These results suggest that the Willems method provides a more reliable estimate of chronological age within a ±1-year margin compared with the other two methods. **Conclusions:** Of the dental age estimators, the Willems method gave the closest age estimates. The Demirjian method overestimated the chronological age in both healthy children and children with FMF. For Turkish children receiving regular colchicine therapy, dental maturation was unaffected by FMF, suggesting that effective inflammatory control might preserve odontogenesis.

## 1. Introduction

Growth represents a multifaceted and dynamic process in which biological systems, particularly the skeletal framework and dental tissues, pass through various stages of development before achieving full maturity [1]. Regarding age estimation, the development of teeth and bones is commonly assessed. Teeth are highly resistant to physical and chemical influences and remain among the last structures to decompose after death, making them reliable indicators for age estimation [2].

The Demirjian method is one of the most popular dental age estimation techniques due to its simplicity, high inter-examiner reliability, and ease of standardization and reproducibility [3,4,5,6]. Since age estimations using the Demirjian method tend to overestimate chronological age, this method was modified and developed by Willems [7,8]. The Cameriere method, on the other hand, is based on measuring the distance between the open apices of the seven left permanent mandibular teeth as seen on panoramic radiographs and by evaluating their relationship with age [9,10,11]. As the accuracy and reliability of dental age estimation methods reported in the literature vary across populations, it is essential that the selected age estimation method is appropriate for the population being studied [12].

Chronic inflammatory diseases are reported to negatively affect bone health, primarily due to the activity of proinflammatory cytokines, such as TNF-α, IL-1, and IL-6, which promote osteoclast-mediated bone resorption and reduce bone mineral density [13,14,15]. Familial Mediterranean fever (FMF) is an inherited autoinflammatory disorder caused by mutations to the MEFV gene, characterized by recurrent febrile episodes and serosal inflammation involving the abdomen, chest, and joints. FMF’s most severe complication is amyloidosis, marked by abnormal protein deposition in organs—particularly the kidneys—that can progress to renal failure [16,17,18]. Colchicine is the primary treatment for FMF, effectively reducing inflammatory attacks and preventing amyloidosis [19]. As therapy is often lifelong, colchicine dose is determined based on age and disease severity. Beyond its anti-inflammatory effects, colchicine can also influence bone biology by disrupting microtubule-mediated processes in osteoblasts, thereby altering matrix secretion and providing cellular protection [20,21].

Age estimation is particularly challenging in individuals with systemic diseases, genetic syndromes, or chronic inflammatory conditions that alter bone development. Such disruptions can accelerate or delay dental and skeletal maturation, thereby reducing the accuracy of conventional estimation methods. Previous studies have highlighted that disorders, such as thalassemia, growth hormone deficiencies, and rheumatic diseases, affect developmental patterns, leading to potential deviations in age assessments. Research focusing on specific patient populations is, therefore, essential for evaluating the reliability and validity of existing methods and minimizing errors in clinical and forensic applications, which is especially crucial in forensic practice where underestimation or overestimation of age can result in serious implications for criminal responsibility. As such, population-specific studies are important for revealing the limitations of current methods and guiding the development of necessary adaptations [22,23].

Recent studies suggest that subclinical inflammation can persist in some patients with FMF, even during symptom-free periods [24,25,26]. In children, this inflammation can lead to loss of appetite and impaired physical and mental development, while its long-term consequences remain insufficiently clarified [27,28]. Despite the high prevalence of FMF in Türkiye, where the disease typically begins during childhood, its potential influence on dental development has not yet been explored [16,17,18]. This study aimed to investigate the possible effects of FMF on dental maturation in the context of its effects on dental development. This study explored three different dental age estimation methods—Demirjian, Willems, and Cameriere methods—to compare healthy children and those with FMF.

## 2. Materials and Methods

In this study, panoramic radiographs of patients were obtained from the Department of Pedodontics, Faculty of Dentistry, Tokat Gaziosmanpaşa University, between 2019 and 2020. A total of 78 children who were diagnosed with FMF based on their medical history, together with 78 systemically healthy patients, were examined, yielding a total sample size of 156 individuals. The study’s ethical approval was obtained from the Local Clinical Research Ethics Committee (ethics number: 19 KAEK 165, approval date: 02.07.2019) and the study was conducted in accordance with the principles set out in the Declaration of Helsinki. All patients (and parents of patients under age 16 years) signed a written informed consent form allowing their data and radiological findings to be used for future research.

### 2.1. Study Design

This retrospective, cross-sectional case–control study was designed in accordance with the Strengthening the Reporting of Observational Studies in Epidemiology (STROBE) guidelines. The control group consisted of children without systemic diseases who were admitted to our clinic during the same time period as the case group. The selection of the control group aimed to achieve a similar distribution of chronological age and gender as in the study group.

The sample size was calculated using G*Power software (version 3.1.9.6; Düsseldorf, Germany), giving a significance level of 5% with a statistical power of 80%, and an effect size of 0.40. A total of 156 participants (78 in each group) were required.

### 2.2. Inclusion Criteria for the FMF Group

Pediatric patients aged 4–15 years;Had no chronic disease in addition to FMF;Regularly use colchicine with a mean dose required to control the disease of 1 mg/day (0.5–2 mg/day);Had no congenitally missing teeth;Had good orthopantomograph quality.

### 2.3. Exclusion Criteria for the Study

Patients who had blurred or distorted radiographs;Had missing teeth (genesis or extraction) in any of the seven left permanent mandibular teeth;Had dental abnormalities (e.g., dilaceration or supernumerary teeth) and developmental disorders (e.g., cleft lip and palate);Had incomplete medical or dental history.

### 2.4. Inclusion Criteria for the Control Group

Pediatric patients aged 4–15 years;Systemically healthy with no diagnosed chronic or autoinflammatory disease;No history or clinical suspicion of FMF and not using colchicine (or other anti-inflammatory or immune-modulating drugs);No congenitally missing teeth;Had high-quality orthopantomographs suitable for evaluation;Had complete medical and dental records.

### 2.5. Exclusion Criteria for the Control Group

Blurred or distorted radiographs;Missing teeth (agenesis or extraction) in any of the seven left permanent mandibular teeth;Dental anomalies (e.g., dilaceration or supernumerary teeth) or developmental disorders (e.g., cleft lip and palate);Incomplete medical or dental history;Any systemic disease diagnosis or regular medication use that could affect growth or dental development.

### 2.6. Radiographic Evaluation

For the Demirjian method of age determination, the mineralization stages of seven teeth in the left mandible were assessed using the mineralization table [3,4,29]. Tables were prepared separately for each age group and sex, with the mineralization stages for each tooth represented by letters A–H. The stages of mineralization were converted into numerical values using sex-specific tables, and the total dental scores of the individuals were established. Dental age was determined by matching the values corresponding to the scores found in the table.

When the mineralization stages of the seven left mandibular teeth were evaluated by the Willems, the mineralization stages A–H of the Demirjian method were taken into account. The scores for each child were marked on the sex-specific tables according to the protocol by Willems et al. [7], and age values calculated.

The root development and the degree of closure of the apical apex of the teeth were assessed using the Cameriere method [9]. The distance between the inner surfaces of the root with an open apex was measured for single-rooted teeth. This measurement was taken separately for each root in multi-rooted teeth. The distance from the tubercle tip to the root apex was measured in these teeth. All these measurements were performed with the ImageJ software (ImageJ 1.46r, NIH, Maryland, MD, USA) at a magnification of ×150, in a dim and quiet environment, with the patient data anonymized. The Cameriere method’s Excel formula was used to convert the measurements to dental age.

Intra-observer reliability was determined by repeating the evaluation of the same radiographs by each researcher after a 2-week interval. Any discrepancies between examiners were resolved with joint re-evaluation and consensus discussion, and if disagreement persisted, a third expert was consulted to ensure consistency across all assessments.

### 2.7. Statistical Analysis

The data were analyzed using R version 4.4.1. The normality was assessed using the Kolmogorov–Smirnov and the Shapiro–Wilk tests. Variables conforming to a normal distribution were compared as pairwise groups using the independent T test. Variables that did not conform to a normal distribution were compared as pairwise groups using the Mann–Whitney U test. Numerical data were represented as mean ± standard deviation and median (minimum–maximum). The interclass correlation coefficient was used to analyze the correlations between the values obtained by each method and age. The significance level was set at *p* < 0.05.

## 3. Results

The distribution of participants by age and sex is presented in Table 1. There were no significant differences between the FMF and control groups regarding chronological or dental age estimates obtained using the Demirjian, Willems, and Cameriere methods (*p* > 0.05 for all comparisons). In the FMF group, dental ages estimated by the Demirjian and Willems methods were slightly higher than chronological ages, whereas dental ages estimated by the Cameriere method were slightly lower (Table 2). However, these differences were not statistically significant (Table 3).

When the differences between dental age and chronological age were compared between the FMF and control groups, no significant difference was found for any estimation methods in the overall sample (*p* > 0.05). Similarly, subgroup analyses based on sex revealed no significant differences between the FMF and control groups (*p* > 0.05 for all comparisons), as shown in Table 4.

Within the narrowest error ranges (±0.25–±0.75), the Cameriere method showed the highest accuracy date rate (54.84%). The Demirjian and Willems methods showed lower performance within this range (approximately 42.58% and 50.97%, respectively). This finding suggests that the Cameriere method provides more precise estimations at smaller tolerance levels (Table 5).

At the ±1-year threshold, the Willems method showed the highest accuracy (69.87%), followed by the Cameriere method (66.67%), while the Demirjian method showed the lowest accuracy (54.49%). These results suggest that Willems’ method provides a more reliable estimate of chronological age within a ±1-year margin compared with the other two methods (Table 5). Findings within the ±1.25–±1.50-year range indicate that the Willems method provides the most reliable results with moderate error tolerances. Intraclass correlation coefficients showed high intra-observer agreement for all three methods in both control and FMF groups (Table 6).

## 4. Discussion

FMF is a chronic autoinflammatory disorder characterized by recurrent episodes of fever and serositis, and results from mutations to the MEFV gene encoding the pyrin protein [30]. The average age of disease onset is approximately 4.5 years. FMF is most prevalent among the Turkish population, with an estimated incidence ranging from 1:150 to 1:1000, and it is reported that nearly 100,000 individuals in Türkiye are affected [17]. To our knowledge, this study is the first to compare the Demirjian, Willems, and Cameriere methods for estimating dental age relative to chronological age in children diagnosed with FMF. Of these methods, the Willems method provided the closest estimation to chronological age in both FMF and healthy children.

Accurate age estimation is of crucial importance for children and adolescents aged 7–21 years in various countries. Under the Turkish penal code, ages 12–15 years represent decisive thresholds for criminal responsibility. Children younger than these ages, including deaf and mute children younger than 15 years, cannot be held legally accountable [22]. In cases where a child’s actual age is uncertain, precise age estimation is indispensable to ensure justice. Errors leading to underestimation or overestimation of age can directly distort sentencing outcomes, resulting in either unjustly lenient or excessively severe penalties. According to Schmeling et al. [23], specific legal age thresholds are of particular importance in forensic age estimation, including ages 14 years (criminal responsibility in some jurisdictions), 16 years (employment or partial legal capacity), 18 years (the internationally recognized age of majority), and 21 years (full legal capacity in specific contexts). In our study, the Demirjian method was less accurate than the other two methods across all age ranges and showed low accuracy within the ±1-year interval, indicating that it was not suitable for the test population.

In our study, the estimated dental ages in the FMF group were slightly higher than chronological ages, although the difference was not significant. This result could be related to the long term use of colchicine, which is the standard therapy for FMF. Beyond its immunomodulatory properties, colchicine has been shown to influence bone metabolism by protecting osteoblasts from inflammatory damage, stabilizing plasma membrane function, and reducing oxidative stress. Through these mechanisms, colchicine not only prevents recurrent inflammatory episodes but could also help maintain bone homeostasis and support normal skeletal and dental development in children with FMF [31]. This protective effect on bone metabolism might explain why dental maturation in patients with FMF receiving regular colchicine therapy appear comparable to, or slightly advanced relative to, healthy children.

Apaydin et al. [32] reported that the Demirjian method overestimated and the Cameriere method underestimated dental age in Turkish children, whereas the Willems method provided the closest estimates to chronological age. In another study from Central Anatolia, the Demirjian method overestimated age by +0.304 years, the Willems method underestimated by –0.06 years, and the Cameriere method underestimated by –0.58 years. Likewise, in Malaysian children, Nolla, Willems, and Demirjian methods tended to overestimate, while Haavikko and Cameriere methods underestimated dental age [33,34]. In our study, the results obtained according to sex were similar to those of the healthy group and were consistent with the existing literature. Both the Demirjian and Willems methods overestimated dental age in healthy and FMF children of both sexes, while the Cameriere method underestimated the age.

In a two-center study conducted in Türkiye with 972 boys and 906 girls aged 5–15 years, the Willems method consistently produced higher dental age estimates than chronological age across all age groups. In contrast, the Cameriere method underestimated dental age in children aged 8–14 years [35]. Hato et al. [33] assessed the accuracy of the Nolla, Willems, and Cameriere methods in 400 children between aged 6–14 years from the Central Black Sea Region of Türkiye. Their results indicated that the Cameriere approach provided the closest approximation to chronological age in both sexes, whereas the Willems method overestimated age by an average of +0.76 years in girls and +0.49 years in boys. Similarly, Koç et al. [36] and Altan et al. [37] observed that the Willems method tended to overestimate dental age in studies on Turkish populations. In line with these findings, our research also showed that the Willems method generally overestimated dental age in the FMF group compared with healthy controls, although it produced more consistent and less biased outcomes.

Altan et al. [38] evaluated the accuracy of the Demirjian and Willems methods in predicting dental age based on chronological age in the Turkish population by examining panoramic radiographs of 745 children aged 4–15.99 years. The Demirjian method overestimated age by 0.832 in females and 0.923 in males, while the Willems method overestimated age by 0.202 in females and 0.434 in males. The Willems method was more accurate than the Demirjian method for the Turkish pediatric population. The Cameriere method was estimated to be lower in both sexes in a study comparing the accuracy of the Cameriere and Willems methods in 636 children aged 6–14.99 years in the Thrace Region [39]. Although there was no statistical difference, the Willems method provided more reliable results with less bias in estimating dental age in Turkish children in our study. The Cameriere method showed an accuracy of 66.7% within ±1 year, with a tendency to underestimate dental age in our study. We consider this underestimation to be related to the method’s fundamental reliance on measuring open apices of developing teeth.

Few studies have examined the effect of systemic disorders on the process of tooth mineralization [40,41,42,43]. In a case–control study, Emeksiz et al. [44] assessed dental age in children with hypothyroidism using the Nolla and Cameriere methods. They retrospectively examined panoramic radiographs of 80 patients with hypothyroidism aged 5–13 years and 80 healthy patients matched for age and sex. In the study, although not significant, the dental age of children with hypothyroidism was slightly lower than that of healthy children by both methods. In another study, a group of 33 adolescents with constitutional growth delay was compared with a healthy group of 41 children. The results showed that chronological age and dental age were similar in children with growth delay [45]. However, a difference was observed when comparing dental age and chronological age with bone age. These findings indicate that bone development is significantly delayed in adolescents with constitutional growth delay, but tooth development progresses in line with their chronological age. In other words, dental maturation is less delayed than bone maturation.

FMF is commonly diagnosed between age 5 and 7 years, a period that is crucial for skeletal and dental development. Colchicine, the first-line treatment for FMF, is routinely used to reduce inflammation and prevent long-term complications, such as amyloidosis [31]. The anti-inflammatory properties of colchicine arise from its ability to block microtubule polymerization, which disrupts neutrophil chemotaxis and inflammatory signaling [46]. Regular and effective use of colchicine might help control inflammation and prevent associated bone destruction. Therefore, inflammation-related bone loss can be reduced in children with FMF, and their mandibular index values could be similar to or even higher than those of healthy children [16]. All patients included in our study were children receiving a mean dose of 1 mg/day (range: 0.5–2 mg/day) of colchicine, which might contribute to the control of systemic inflammation and the preservation of dental immaturity.

The main limitations of our study were the inability to apply the methods to children with missing teeth and its single-center design. In addition, the retrospective nature of the study might limit the generalizability of the findings and restrict control over potential confounding variables. However, an important strength of our research is the inclusion of many cases, which provide valuable data contributing to the literature on the dental development of children with FMF. Future studies might benefit from using methods that investigate the relationship between bone age and dental age, offering a more comprehensive understanding of developmental patterns in this patient group. In addition, considering different drug dosages, varying follow-up periods for the disease, and comparing groups with and without medication use in future studies would further enhance the scope and clinical value of the findings.

## 5. Conclusions

It is essential to understand the effects of diseases that affect the growth and development of hard tissue. In our study, in Turkish children receiving regular colchicine therapy, dental maturation appears unaffected by FMF, suggesting that effective inflammatory control can preserve odontogenesis. These findings indicate that the Willems and Cameriere methods provide more accurate age estimations compared with the Demirjian method. In both healthy children and children with FMF, the Demirjian method overestimated chronological age. Regular and effective use of colchicine could help control inflammation and prevent associated bone and dental immaturity.

## Figures and Tables

**Table 1 children-12-01458-t001:** Distribution of patients with familial Mediterranean fever and the control group according to age.

Groups	Chronological Age (Years)										
	5–5.99	6–6.99	7–7.99	8–8.99	9–9.99	10–10.99	11–11.99	12–12.99	13–13.99	14–14.99	15–15.99
Control											
Girls	2	3	1	5	1	8	3	5	4	3	0
Boys	3	3	1	9	6	7	4	3	7	0	0
FMF											
Girls	2	4	3	1	4	4	5	3	4	3	0
Boys	3	3	4	4	6	5	6	4	2	6	2

**Table 2 children-12-01458-t002:** Comparison of chronological age and dental age values estimated by the Demirjian, Willems, and Cameriere methods according to gender.

	Girl (n = 68)		Boy (n = 88)		Total (n = 176)		Test Statics	*p*
	Mean ± S.D	Med (min–max)	Mean ± S.D	Med (min–max)	Mean ± S.D	Med (min–max)		
Chronologic Age	10.4 ± 2.65	10.5 (5.40–14.8)	10.21 ± 2.68	10.15 (5.00–15.8)	10.3 ± 2.66	10.3 (5–15.8)	0.510	0.611
Demirjian	11.5 ± 2.62	12.0 (6.80–16.0)	10.98 ± 2.81	10.40 (5.50–16.0)	11.2 ± 2.73	11.4 (5.5–16.0)	1.159	0.248
Willems	10.9 ± 2.53	11.8 (5.72–16.0)	10.54 ± 2.60	10.68 (5.21–16.0)	10.7 ± 2.57	11 (5.21–16.0)	0.929	0.354
Cameriere	10.1 ± 2.23	10.4 (5.60–16.0)	9.46 ± 2.41	9.46 (5.68–16.0)	9.94 ± 2.33	10.3 (5.60–16.0)	0.929	0.354

Independent Samples T-Test.

**Table 3 children-12-01458-t003:** Comparison of chronological age and dental age values estimated by the Demirjian, Willems, and Cameriere methods between the FMF and control groups.

	Control (n = 78)		FMF (n = 78)		Test Statistic	*p*
	Mean ± S.D	Med (min–max)	Mean ± S.D	Med (min–max)		
Chronologic Age	10.14 ± 2.55	10.1 (5.20–14.5)	10.5 ± 2.77	10.6 (5.0–15.8)	−0.764	0.446
Demirjian	11.10 ± 2.63	11.7 (6.80–16.0)	11.30 ± 2.85	10.3 (5.50–16.0)	−0.437	0.663
Willems	10.63 ± 2.34	11.3 (6.09–14.3)	10.8 ± 2.80	10.9 (5.21–16.0)	−0.383	0.703
Cameriere	9.73 ± 2.0	10.2 (6.32–13.3)	10.1 ± 2.62	10.4 (5.60–16.0)	−1.123	0.263

Independent Samples T-Test.

**Table 4 children-12-01458-t004:** Comparison of dental age differences (CA–DA) by method and group.

Gender		Control	FMF	*p*
Total	DA-CAd	0.96 ± 0.99	0.83 ± 0.95	0.496 ^x^
	DA-CAw	0.48 ± 0.80	0.32 ± 0.93	0.255 ^x^
	DA-CAc	−0.41± 0.94	−0.31± 0.94	0.718 ^y^
Girl	DA-CAd	1.18 ± 1.18	0.94 ± 0.79	0.064 ^x^
	DA-CAw	0.65 ± 0.95	0.33 ± 0.94	0.260 ^x^
	DA-CAc	−0.41 ± 1.08	−0.16 ± 1.08	0.415 ^y^
Boy	DA-CAd	0.79± 0.79	0.75 ± 1.06	0.595 ^x^
	DA-CAw	0.35 ± 0.65	0.32 ± 1.02	0.835 ^x^
	DA-CAc	−0.41 ± 0.83	−0.43 ± 1.10	0.863 ^x^

x Independent T-Test; y Mann–Whitney U Test, *p* < 0.05, DA: Dental age, CA: Chronologic Age, d = Demirjian method, w = Willems method, c = Cameriere method, FMF: Familial Mediterranean Fever.

**Table 5 children-12-01458-t005:** Accuracy rates of dental age predictions based on different estimation methods.

	Demirjian	Willems	Cameriere
±0.25	18.06	18.06	14.84
±0.50	33.55	30.97	38.06
±0.75	42.58	50.97	54.84
**±1.00**	**54.49**	**69.87**	**66.67**
±1.25	62.58	78.71	80.00
±1.50	72.90	89.03	83.87
±1.75	80.00	92.26	88.39
±2.00	90.97	98.71	94.84

**Table 6 children-12-01458-t006:** Evaluation of intra-observer agreement for three methods.

Gender	Group	ICC (%95 CI) ^a^	ICC (%95 CI) ^b^	ICC (%95 CI) ^c^
Total	Control	0.942 (0.912–0.961)	0.960 (0.939–0.973)	0.943 (0.914–0.962)
	FMF	0.971 (0.954–0.981)	0.971 (0.955–0.982)	0.962 (0.940–0.976)
Girl	Control	0.926 (0.863–0.960)	0.957 (0.920–0.977)	0.942 (0.893–0.969)
	FMF	0.958 (0.918–0.979)	0.977 (0.954–0.989)	0.967 (0.934–0.984)
Boy	Control	0.959 (0.927–0.976)	0.961 (0.932–0.978)	0.940 (0.895–0.966)
	FMF	0.966 (0.938–0.981)	0.967 (0.940–0.982)	0.962 (0.926–0.978)

ICC: Intraclass correlation coefficient; ^a^ Demirjian; ^b^ Willems; ^c^ Cameriere, FMF: Familial Mediterranean Fever.

## Data Availability

The datasets analyzed during this study are available from the corresponding author on reasonable request. The data are not publicly available due to privacy and ethical restrictions.

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
