# Peer review of "Dental Age Estimation by Demirjian, Willems, and Cameriere Methods in Children with Familial Mediterranean Fever: A Case–Control Study"

_children, 2025, doi:10.3390/children12111458_

Round 1
Reviewer 1 Report
Comments and Suggestions for Authors
This manuscript needs some more attention as the followings:
- explain more clearly why FMF could affect dental maturation.
- Acknowledge and justify small sample size vs the power of the findings.
- avoid speculation, and expand on the clinical/forensic significance.
- polish grammar, shorten long sentences, streamline the abstract, and cut redundancies.
Author Response
Dear Editor and Reviewers,
We would like to thank you very much for the valuable time and constructive comments provided. We have carefully revised the manuscript according to the reviewers’ suggestions. Below, we provide a point-by-point response to each comment.
Comments 1: Explain more clearly why FMF could affect dental maturation.
Response 1: We revised the Introduction to clarify the potential link between systemic inflammatory diseases and growth/development. We highlighted that FMF, due to recurrent inflammation and cytokine release, may influence skeletal maturation, although its effects on dental development remain unclear.
“Chronic inflammatory diseases are well-documented to negatively affect bone health, primarily due to the activity of pro-inflammatory cytokines such as TNF-α, IL-1, and IL-6, which promote osteoclast-mediated bone resorption and reduce bone mineral density. Familial Mediterranean Fever (FMF), an inherited autoinflammatory disorder caused by mutations in the MEFV gene, is characterized by recurrent febrile episodes and serosal inflammation involving the abdomen, chest, and joints. Its most severe complication is amyloidosis, marked by abnormal protein deposition in organs—particularly the kidneys—that may progress to renal failure. [12-14] Colchicine is the primary treatment for FMF, effectively reducing inflammatory attacks and preventing amyloidosis. [15] As therapy is generally lifelong, the dosage is tailored to age and disease severity. Beyond its anti-inflammatory effects, colchicine may also influence bone biology by disrupting microtubule-mediated processes in osteoblasts, thereby altering matrix secretion and providing cellular protection [16,17]”
(Page 2, Introduction, lines 75–87)
Comments 2: Acknowledge and justify small sample size vs the power of the findings.
Response 2: We acknowledged the limitation of our relatively small sample size in the Discussion. The power of study was “In this study, the primary comparison was made between the mean chronologic age of the control and FMF groups. With a power of 80%, a significance level of 5%, and an effect size of 0.40, a total of 156 participants (78 in each group) were included. The power analysis was performed using the G*Power software (version 3.1.9.6).”
(Page 4, Methods, lines 170-174)
Comments 3: Avoid speculation, and expand on the clinical/forensic significance.
Response 3: Speculative statements were reduced. We expanded on the implications of our findings, particularly the relevance for forensic dental age estimation and medicine.
We added “Age determination is particularly challenging in individuals with systemic diseases, genetic syndromes, or chronic inflammatory conditions that alter bone development. Such disruptions can accelerate or delay dental and skeletal maturation, thereby reducing the accuracy of conventional estimation methods. Previous studies have highlighted that disorders such as thalassemia, growth hormone deficiencies, and rheumatic diseases significantly affect developmental patterns, leading to potential deviations in age assessments. Research focusing on specific patient populations is, therefore, essential for evaluating the reliability and validity of existing methods and minimizing errors in clinical and forensic applications. This is especially critical in forensic practice, where underestimation or overestimation of age may result in serious implications for criminal responsibility. In this regard, population-specific studies make an important scientific contribution by revealing the limitations of current methods and guiding the development of necessary adaptations [18, 19].”
(Page 2, Introduciton, lines 83-101)
- Coşkun N; Koç A; Koç E; Karakaş HM. Forensic age estimation: A study on Turkish children using Demirjian standards. Forensic Sci Int. 2010;194:46.e1–46.e4. https://doi.org/10.1016/j.forsciint.2009.10.015
19.Schmeling A; Grundmann C; Fuhrmann A; et al. Criteria for age estimation in living individuals. Int J Legal Med. 2008;122(6):457–460. https://doi.org/10.1007/s00414-008-0254-2
“Accurate age estimation is of critical importance for children and adolescents between the ages of 7 and 21 in various countries. Under the Turkish Penal Code, the ages of 12 and 15 represent decisive thresholds for criminal responsibility. Children below these ages, including deaf and mute children under 15, cannot be held legally accountable [18]. In cases where a child's actual age is uncertain, precise age estimation is indispensable to ensure justice. Errors leading to underestimation or overestimation of age may directly distort sentencing outcomes, resulting in either unjustly lenient or excessively severe penalties. According to Schmeling et al. [19], certain legal age thresholds are of particular importance in forensic age estimation, including 14 years (criminal responsibility in some jurisdictions), 16 years (employment or partial legal capacity), 18 years (the internationally recognized age of majority), and 21 years (full legal capacity in specific contexts).”
(Page 7, Discussion, lines: 262–273)
[18] Coşkun N; Koç A; Koç E; Karakaş HM. Forensic age estimation: A study on Turkish children using Demirjian standards. Forensic Sci Int. 2010;194:46.e1–46.e4. https://doi.org/10.1016/j.forsciint.2009.10.015
[19]Schmeling A; Grundmann C; Fuhrmann A; et al. Criteria for age estimation in living individuals. Int J Legal Med. 2008;122(6):457–460. https://doi.org/10.1007/s00414-008-0254-2
Comments 4: Polish grammar, shorten long sentences, streamline the abstract, and cut redundancies.
Response 4: The abstract was restructured to be more concise and informative (page 1). Redundant sentences were removed throughout the text. Language was revised for clarity and style.
(Page 1, Abstract, lines 34-53)
Reviewer 2 Report
Comments and Suggestions for Authors This study concerns the application of several common dental age estimation methods to a sample of patients with Familial Mediterranean Fever. The authors reported the absence of other studies on the topic, and indeed, similar studies seem to be absent in the literature. Age determination can be challenging in cases of pathologies affecting bone development; therefore, in my opinion, it is important to report studies on specific populations. However, I believe the study needs to be revised, particularly regarding the existing literature on the various dental age estimation methods. Other comparative studies and reviews exist, but they are not considered by the authors. I believe it is essential to thoroughly analyze the existing literature to fully address the issue. The same applies to the introduction, which should include a more thorough description of the methods applied in the study and their limitations. Finally, there are some typo to fix.Author Response
Dear Editor and Reviewers,
We would like to thank you very much for the valuable time and constructive comments provided. We have carefully revised the manuscript according to the reviewers’ suggestions. Below, we provide a point-by-point response to each comment.
Comments 1: The existing literature on dental age estimation methods was not fully considered.
Response 1: We substantially enriched both the Introduction and Discussion with recent meta-analyses, comparative studies, and reviews on Demirjian, Willems, and Cameriere methods (pages 2-4 and 8–11). For example, we cited Cureus 2024, Int J Legal Med 2025, and Healthcare 2021 meta-analysis.
[6] Al-Juhani A; Binshalhoub A; Showail S; et al. Comparative analysis of dental age estimation: A systematic review and meta-analysis assessing gender-specific accuracy of the Demirjian and Nolla methods across different age groups. Cureus. 2024;16(12):e75031. https://doi.org/10.7759/cureus.75031
[8] Brasil DDCN; Moreira DD; Santiago BM; Vieira WA; Avdeenko O; Paranhos LR; Franco A. Global lens of Willems' method for dental age estimation: where we are and where we are going – umbrella review. Int J Legal Med. 2025;139(3):1183–1192. https://doi.org/10.1007/s00414-025-03424-2
[11] Hostiuc S; Diaconescu I; Rusu MC; Negoi I. Age estimation using the Cameriere methods of open apices: A meta-analysis. Healthcare. 2021;9:237. https://doi.org/10.3390/healthcare9020237
Comments 2: Provide a more thorough description of the methods and their limitations.
Response 2: “The main limitations of our study are the inability to apply the methods to children with missing teeth and its single-centre design. However, an important strength of our research is the inclusion of a large number of cases, which provides valuable data contributing to the literature on the dental development of children with FMF. Future studies may benefit from using methods that investigate the relationship between bone age and dental age, offering a more comprehensive understanding of developmental patterns in this patient group. In addition, considering different drug dosages, varying follow-up periods for the disease, and comparing groups with and without medication use in future studies would further enhance the scope and clinical value of the findings.”
(Page 9, Discussion, lines 371–379)
Comments 3:Typographical errors.
Response 3: All typographical and formatting issues have been corrected.
Reviewer 3 Report
Comments and Suggestions for Authors
The manuscript addresses an interesting and clinically relevant topic, namely the potential effects of Familial Mediterranean Fever (FMF) on dental age estimation using three widely accepted methods. To my knowledge, this is the first study of its kind, which gives the work originality and potential significance. The Introduction is well written and provides a comprehensive background with relevant literature, although the flow can be improved by shortening repetitive sections and ensuring a sharper focus on the knowledge gap. Some sentences are overly descriptive and could be streamlined to make the research aim stand out more clearly.
The research design, a retrospective case–control study with 78 FMF patients and 78 matched controls, is appropriate in principle. However, several methodological details require clarification. For example, the selection process of controls is not fully explained, and it is unclear whether they were strictly age- and sex-matched. The inclusion and exclusion criteria are clearly stated, but the exact dosage description of colchicine as “1 mg/kg” seems unusually high and should be checked for accuracy, as colchicine is typically prescribed at lower doses in children. In addition, inter- and intra-examiner reliability are mentioned, but more detail on the calibration process of the two examiners would strengthen the methodological rigor.
The Results are well structured and tables are clearly presented, allowing the reader to follow the data without difficulty. However, the text sometimes repeats numerical values already given in the tables, which could be reduced to improve readability. A more critical interpretation of effect sizes and clinical implications, beyond the statement that no significant difference was found, would add depth. For example, what does a small but consistent under- or overestimation by a given method mean for clinical practice or forensic applications?
The Discussion provides a broad literature context, yet it occasionally drifts into long summaries of previous studies without sufficiently contrasting them with the present findings. The authors should more strongly emphasize why their study adds value, especially given that systemic diseases often alter skeletal development, but here no significant effect was found on dental development. A more nuanced reflection on the potential protective role of colchicine is welcome but should be phrased cautiously, as causality cannot be established. The limitations are briefly acknowledged, but the retrospective design, relatively small sample size, and single-center setting deserve stronger emphasis. It would also be valuable to suggest concrete directions for future research, such as longitudinal follow-up or inclusion of untreated FMF patients.
The English is generally understandable but requires editing for clarity, conciseness, and grammar. Some sentences are long and awkwardly phrased, which slightly hinders readability. I recommend professional language editing before publication.
Author Response
Dear Editor and Reviewers,
We would like to thank you very much for the valuable time and constructive comments provided. We have carefully revised the manuscript according to the reviewers’ suggestions. Below, we provide a point-by-point response to each comment.
Comments 1: The flow of the Introduction could be improved by shortening repetitive sections.
Response 1: We revised the Introduction to reduce repetition and sharpen the focus on the knowledge gap.
Comments 2: Clarify the selection of controls and age/sex matching.
Response 2: This retrospective, cross-sectional case–control study was designed in accordance with the Strengthening the Reporting of Observational Studies in Epidemiology (STROBE) guidelines. The control group consisted of children without systemic diseases who were admitted to our clinic during the same time period as the case group. The selection of the control group aimed to achieve a similar distribution of chronological age and gender as in the study group. (Page 3, Methods, lines 125–130)
Comments 3: The colchicine dosage of 1 mg/kg seems unusually high and should be checked.
Response 3: This was an oversight. We corrected the dosage to reflect the clinically used range (1 mg/day) in children. (Page 3, Methods, lines 134). We sincerely thank the reviewer for identifying this important error. We apologize for the oversight and have corrected the dosage in the revised manuscript to reflect the clinically appropriate range used in pediatric practice
Comments 4: Provide more detail on examiner calibration.
Response 4: We added details on calibration: Intraobserver reliability was determined by repeating the evaluation of the same radiographs by each researcher after a two-week interval.
(Page 4, Methods, lines 167-168)
Comments 5: Results repeat numbers from tables.
Response 5: We streamlined the Results section to avoid repeating numerical values already shown in tables. (Page 4-6)
Comments 6: Interpret effect sizes and clinical implications more critically.
Response 6: We expanded the Discussion to explain that while no significant differences were observed, small consistent trends across methods may have implications in borderline forensic cases.
“In our study, dental ages in the FMF group were slightly higher than chronological ages, although the difference did not reach statistical significance. This result may be related to the long-term use of colchicine, the standard therapy for FMF. Beyond its immunomodulatory properties, colchicine has been shown to influence bone metabolism by protecting osteoblasts from inflammatory damage, stabilizing plasma membrane function, and reducing oxidative stress. Through these mechanisms, colchicine not only prevents recurrent inflammatory episodes but may also help maintain bone homeostasis and support normal skeletal and dental development in children with FMF [27]. This protective effect on bone metabolism may explain why dental maturation in FMF patients receiving regular colchicine therapy appears comparable to, or slightly advanced relative to, their healthy peers.”
(Page 7, Discussion, lines 275–285)
Within ±1.00 year, Willems (69.9%) and Cameriere (66.7%) outperformed Demirjian (54.5%), indicating superior precision.
The Cameriere method demonstrated an accuracy rate of 66.7% within ±1 year, with a tendency to underestimate dental age in our study. We consider this underestimation to be related to the method’s fundamental reliance on measuring open apices of developing teeth.
Comments 7: Strengthen limitations and suggest future research.
Response 7: We emphasized “The main limitations of our study are the inability to apply the methods to children with missing teeth and its single-centre design. However, an important strength of our research is the inclusion of a large number of cases, which provides valuable data contributing to the literature on the dental development of children with FMF. Future studies may benefit from using methods that investigate the relationship between bone age and dental age, offering a more comprehensive understanding of developmental patterns in this patient group. In addition, considering different drug dosages, varying follow-up periods for the disease, and comparing groups with and without medication use in future studies would further enhance the scope and clinical value of the findings.”
(Page 9, Discussion, lines 371–379)
Comments 8: English needs editing.
Response 8: We performed thorough language editing throughout the manuscript.
Reviewer 4 Report
Comments and Suggestions for Authors
Dear Authors,
The aim of this study was to investigate the possible impact of familial Mediterranean fever (FMF) on dental development by comparing the dental age of sick and generally healthy children using three different methods of estimating dental age.
The research tools used in this study were accurate. This applies to both the selection and exclusion of study groups, the assessed indicators, and the statistical analysis. This allowed for a thorough and detailed achievement of the research objective.
The content of the article provides interesting and valuable scientific information, particularly that of the research conducted by the first author.
The strengths of this work include: a reliable assessment of the accuracy of individual dental age assessment methods – important for scientific research and clinically useful; confirmation that no effect of the disease on dental age was observed in children with FMF – very important information (medicine-based evidence).
The weaknesses of the manuscript include: Tables 1 and 2 are difficult to read – the tables overlap and are difficult to analyze. Additionally: see lines 298 – 29 and 234 – 25 – ???
The reviewer believes that it would be worthwhile to examine the impact of disease duration and colchicine treatment on other components of the dentition in the future.
Author Response
Dear Editor and Reviewers,
We would like to thank you very much for the valuable time and constructive comments provided. We have carefully revised the manuscript according to the reviewers’ suggestions. Below, we provide a point-by-point response to each comment.
Comments 1: Tables 1 and 2 are difficult to read.
Response 1: Tables 1 and 2 were reformatted to improve readability and prevent overlap (pages 5).
Comments 2: Clarify references in relevant lines (298–29 and 234–25).
Response 2: References in relevant lines have been corrected.
Comments 3: Future research on disease duration and colchicine treatment.
Response 3: We sincerely thank the reviewer for this valuable suggestion. We have acknowledged this point in the Discussion and emphasized that future research should investigate the influence of disease duration and colchicine treatment on different aspects of dental development.“Future studies may benefit from using methods that investigate the relationship between bone age and dental age, offering a more comprehensive understanding of developmental patterns in this patient group. In addition, considering different drug dosages, varying follow-up periods for the disease, and comparing groups with and without medication use in future studies would further enhance the scope and clinical value of the findings.”
(Page 9, Discussion, lines 374–379)
Round 2
Reviewer 1 Report
Comments and Suggestions for Authors
Refine the conclusion to: In Turkish children under regular colchicine therapy, dental maturation appears unaffected by Familial Mediterranean Fever, suggesting effective inflammatory control may preserve odontogenesis.
Author Response
Dear Editor and Reviewers,
We would like to thank you very much for the valuable time and constructive comments provided. We have carefully revised the manuscript according to the reviewers’ suggestions. Below, we provide a point-by-point response to each comment.
Comment 1
Refine the conclusion to: “In Turkish children under regular colchicine therapy, dental maturation appears unaffected by Familial Mediterranean Fever, suggesting effective inflammatory control may preserve odontogenesis.”
Response 1
We thank the reviewer for this valuable suggestion. The proposed refinement has been added to the conclusion section to improve clarity and clinical relevance.
(380-382 line, page 9)
Reviewer 2 Report
Comments and Suggestions for Authors Although the review suggestions were considered, I remain uncertain about the literature included and, consequently, the authors' actual background. The most prominent authors in the fields of anthropology and age estimation were not considered at all, or were only poorly considered. This is important not only for citation purposes but especially for the existing background and the gaps in the literature that the manuscript aims to address.Author Response
Dear Editor and Reviewers,
We would like to thank you very much for the valuable time and constructive comments provided. We have carefully revised the manuscript according to the reviewers’ suggestions. Below, we provide a point-by-point response to each comment.
Comment 1
The most prominent authors in the fields of anthropology and age estimation were not considered at all, or were only poorly considered. This is important not only for citation purposes but especially for the existing background and the gaps in the literature that the manuscript aims to address.
Response 1
We sincerely thank the reviewer for this thoughtful comment. We carefully reviewed the relevant literature and aimed to include the most pertinent studies related to our research scope. However, we acknowledge that additional references from leading authors in the fields of anthropology and age estimation could further strengthen the manuscript. We would be pleased to incorporate any specific references the reviewer recommends to enhance the scientific depth and contextual accuracy of our work. In this regard, to strengthen the theoretical background and ensure that the key literature is appropriately represented, the works of Helen M. Liversidge (2024) have been added to the Introduction section. (page 2, line 75)
References added:
Liversidge, H. M. (2024). Commentary on “new systems for dental maturity based on seven and four teeth” Demirjian and Goldstein, Annals of Human Biology, 1976, 3, 411–421. Annals of Human Biology, 51(1). https://doi.org/10.1080/03014460.2024.2401026
Reviewer 3 Report
Comments and Suggestions for Authors
The authors have made substantial improvements in response to the initial review, particularly by clarifying several methodological details, tightening the Introduction, and better structuring the Discussion. The overall manuscript is now more coherent and easier to follow. Nevertheless, a few issues still require attention before final acceptance:
Methods:
-
The inclusion criteria now state that colchicine dosage was 1 mg/day, which is clearer than the previous “1 mg/kg.” This correction is important, but it would be useful to briefly explain whether dosage was standardized across all patients or adjusted for age/weight.
-
The description of examiner calibration and intra-/inter-observer reliability has been expanded, which is positive. However, a short statement on how disagreements (if any) were resolved would further strengthen transparency.
Results:
-
Tables are comprehensive and clear. However, some repetition of numerical data in the text remains. Summarizing the key findings rather than restating full numbers would improve readability.
-
The accuracy rates (Table 5) are very informative and should be emphasized more strongly in the Results or Discussion, as they clearly demonstrate the relative performance of the methods.
Discussion:
-
The section is more focused than in the original version, but it still contains long summaries of other studies. Consider condensing these paragraphs and directing the emphasis toward how your findings specifically add to the literature on systemic disease and dental development.
-
The hypothesis of colchicine’s protective role is interesting and is now presented more cautiously, but the wording could be further refined to avoid implying causality. For example, phrases like “may help maintain” or “might contribute” are preferable.
-
Limitations are more clearly acknowledged, but please explicitly state that the retrospective design and single-center setting restrict generalizability.
The English is understandable but still requires editing for conciseness and grammar. Sentences are sometimes long and could be broken down for clarity. For example, the Discussion paragraphs on colchicine’s potential effects could be streamlined considerably.
Author Response
Dear Editor and Reviewers,
We would like to thank you very much for the valuable time and constructive comments provided. We have carefully revised the manuscript according to the reviewers’ suggestions. Below, we provide a point-by-point response to each comment.
Comment 1
The inclusion and exclusion criteria are clearly stated, but the exact dosage description of colchicine as “1 mg/kg” seems unusually high and should be checked for accuracy, as colchicine is typically prescribed at lower doses in children.
Response 1
We thank the reviewer for this helpful comment. In the Materials and Methods and Discussion sections, the previous “1 mg/kg” dosage was corrected to: “Regularly use colchicine (The mean colchicine dose required to control the disease was 1 mg/day (0.5–2 mg/day)).” In our study, the colchicine dosage was standardized at 1 mg/day for all patients, reflecting the routine clinical dosage prescribed by pediatric rheumatologists in our institution.
(141-142 lines, page 3 / 364-366 lines, page 9 )
Comment 2
The description of examiner calibration and intra-/inter-observer reliability has been expanded, which is positive. However, a short statement on how disagreements (if any) were resolved would further strengthen transparency.
Response 2:
This sentence was added “ Any discrepancies between examiners were resolved through joint re-evaluation and consensus discussion, and if disagreement persisted, a third expert was consulted to ensure consistency across all assessments.”
(176-179 lines, page 4)
Comment 3
Tables are comprehensive and clear. However, some repetition of numerical data in the text remains. Summarizing the key findings rather than restating full numbers would improve readability.
Response 3
This section was corrected as this.
“The distribution of participants by age and gender is presented in Table 1.
No statistically significant differences were found between girls and boys in terms of either chronological or dental age estimates obtained using the Demirjian, Willems, and Cameriere methods (p > 0.05 for all comparisons). In particular, the Demirjian method showed a tendency for higher mean dental age values in girls compared with boys, although this difference did not reach statistical significance (Table 2).
There were no statistically significant differences were observed between the FMF and control groups regarding chronological or dental age estimates obtained by the Demirjian, Willems, and Cameriere methods (p > 0.05 for all comparisons). The FMF group showed a slight tendency toward higher mean chronological and dental ages compared with the control group; however, these differences were not statistically significant (Table 3)."
(Pages 3–4, Results section)
Comment 4
The accuracy rates (Table 5) are very informative and should be emphasized more strongly in the Results or Discussion, as they clearly demonstrate the relative performance of the methods.
Response 4
We thank the reviewer for this valuable comment. In line with the suggestion, the accuracy rates presented in Table 5 have been emphasized more clearly in the Results section. Additional explanatory paragraphs were added to provide a more detailed interpretation of the accuracy findings, highlighting the comparative performance of the Demirjian, Willems, and Cameriere methods across different error ranges.
"In the narrowest error ranges (±0.25–±0.75), the Cameriere method demonstrated the highest accuracy rate (54.84%). The Demirjian and Willems methods showed lower performance within this range (approximately 42.58% and 50.97%, respectively). This finding suggests that the Cameriere method provides more precise estimations at smaller tolerance levels (Table 5)." (page 5)
"At the ±1.0 year threshold, Willems method demonstrated the highest accuracy (69.87%), followed closely by Cameriere method (66.67%), while Demirjian method showed the lowest accuracy (54.49%). These results suggest that Willems' method provides a more reliable estimate of chronological age within a ±1.0 year margin compared to the other two methods (Table 5). Findings within the ±1.25–±1.50 range indicate that the Willems method provides the most reliable results at moderate error tolerances." (page 5)
Comment 5
The section is more focused than in the original version, but it still contains long summaries of other studies. Consider condensing these paragraphs and directing the emphasis toward how your findings specifically add to the literature on systemic disease and dental development.
Response 5
We thank the reviewer for this constructive comment. In accordance with the suggestion, the Discussion section has been revised to be more concise and to emphasize how our findings contribute to the literature on systemic disease and dental development. Additional comparative explanations have been added to contextualize our results.
"In our study, the Demirjian method performed worse than the other two across all ranges and showed low accuracy within the ±1.00-year interval, suggesting it was not suitable for this population. "( page 7, lines 284-287)
"Apaydın et al. [29] reported that Demirjian overestimated and Cameriere underestimated dental age in Turkish children, whereas Willems provided the closest estimates to chronological age. Similarly, in another study from Central Anatolia, Demirjian overestimated age by +0.304 years, Willems by –0.06 years, and Cameriere by –0.58 years. In Malaysian children, Nolla, Willems, and Demirjian tended to overestimate, while Haavikko and Cameriere underestimated dental age [30, 31]. In our study evaluating dental age estimation methods in children with FMF, the results obtained by gender were consistent with those of the healthy group and aligned with the existing literature. Both Demirjian and Willems methods overestimated dental age in healthy and FMF children of both sexes, while the Cameriere method underestimated it." (page 8, lines 300-309)
This revision refines the Discussion section by reducing redundancy and clearly highlighting how the present findings support and extend current knowledge.
Comment 6
The hypothesis of colchicine’s protective role is interesting and is now presented more cautiously, but the wording could be further refined to avoid implying causality. For example, phrases like “may help maintain” or “might contribute” are preferable.
Response 6
We thank the reviewer for this valuable comment. The wording in the manuscript has been revised to avoid implying causality and to reflect a more cautious interpretation.
Revised sentence:
“All patients included in our study were children receiving regular colchicine treatment at a dosage of 1 mg/day, which might contribute to the control of systemic inflammation and the preservation of dental maturation.” (364-366 line page 9)
Comment 7
Limitations are more clearly acknowledged, but please explicitly state that the retrospective design and single-center setting restrict generalizability.
Response 7
We thank the reviewer for this helpful recommendation. The Limitations section has been updated to explicitly mention that the retrospective design and single-center setting may limit the generalizability of the study findings.
(368-370 lines, page 9)